

# Sarc-F and muscle function in community dwelling adults with aged care service needs: baseline and post-training relationship

Justin W.L. Keogh[1,2,3,4], Tim Henwood[1,5,6], Paul A. Gardiner[7,8], Anthony G. Tuckett[9,10], Sharon Hetherington[11], Kevin Rouse[11] and Paul Swinton[12]

[1] Faculty of Health Sciences and Medicine, Bond University, Robina, Queensland, Australia
[2] Human Potential Centre, Auckland University of Technology, Auckland, New Zealand
[3] Kasturba Medical College, Manipal Academy of Higher Education, Mangalore, Karnataka, India
[4] Cluster for Health Improvement, Faculty of Science, Health, Education and Engineering, University of the Sunshine Coast, Sunshine Coast, Queensland, Australia
[5] Current affiliation: Southern Cross Care SA and NT, Adelaide, Australia
[6] School of Human Movement and Nutritional Science, University of Queensland, Brisbane, Australia
[7] Faculty of Medicine, University of Queensland, Brisbane, Australia
[8] Mater Research Institute, University of Queensland, Brisbane, Australia
[9] School of Nursing, Midwifery and Social Work, University of Queensland, Brisbane, Australia
[10] College of Nursing, Yonsei University, Seoul, South Korea
[11] The Chermside Senior Citizens Centre, Burnie Brae, Brisbane, Australia
[12] School of Health Sciences, Robert Gordon University, Aberdeen, Scotland

Corresponding author
Justin W.L. Keogh,
jkeogh@bond.edu.au

## ABSTRACT

**Background**. This study sought to better understand the psychometric properties of the SARC-F, by examining the baseline and training-related relationships between the five SARC-F items and objective measures of muscle function. Each of the five items of the SARC-F are scored from 0 to 2, with total score of four or more indicative of likely sarcopenia.

**Methods**. This manuscript describes a sub-study of a larger step-wedge, randomised controlled 24-week progressive resistance and balance training (PRBT) program trial for Australian community dwelling older adults accessing government supported aged care. Muscle function was assessed using handgrip strength, isometric knee extension, 5-time repeated chair stand and walking speed over 4 m. Associations within and between SARC-F categories and muscle function were assessed using multiple correspondence analysis (MCA) and multinomial regression, respectively.

**Results**. Significant associations were identified at baseline between SARC-F total score and measures of lower-body muscle function ($r = -0.62$ to $0.57$; $p \leq 0.002$) in 245 older adults. MCA analysis indicated the first three dimensions of the SARC-F data explained 48.5% of the cumulative variance. The initial dimension represented overall sarcopenia diagnosis, Dimension 2 the ability to displace the body vertically, and Dimension 3 walking ability and falls status. The majority of the 168 older adults who completed the PRBT program reported no change in their SARC-F diagnosis or individual item scores (56.5–79.2%). However, significant associations were obtained between training-related changes in SARC-F total and item scores and changes in walking speed and chair
stand test performance ($r = -0.30$ to $0.33$; $p < 0.001$ and relative risk ratio $= 0.40$–$2.24$; $p < 0.05$, respectively). MCA analysis of the change score data indicated that the first two dimensions explained 32.2% of the cumulative variance, with these dimensions representing whether a change occurred and the direction of change, respectively.

**Discussion**. The results advance our comprehension of the psychometric properties on the SARC-F, particularly its potential use in assessing changes in muscle function. Older adults' perception of their baseline and training-related changes in their function, as self-reported by the SARC-F, closely matched objectively measured muscle function tests. This is important as there may be a lack of concordance between self-reported and clinician-measured assessments of older adults' muscle function. However, the SARC-F has a relative lack of sensitivity to detecting training-related changes, even over a period of 24 weeks.

**Conclusions**. Results of this study may provide clinicians and researchers a greater understanding of how they may use the SARC-F and its potential limitations. Future studies may wish to further examine the SARC-F's sensitivity of change, perhaps by adding a few additional items or an additional category of performance to each item.

# INTRODUCTION

The SARC-F is a quick self-report sarcopenia screening tool involving five questions (items) assessing an older individual's muscular strength, ability to walk, rise from a chair and climb stairs as well as fall status (*Malmstrom & Morley, 2013*). It is endorsed for use in clinical practice and research by the European Working Group Sarcopenia in Older People (EWGSOP2) consensus (*Cruz-Jentoft et al., 2019*).

Previous research indicates that the SARC-F can predict quality of life, physical limitation, hospitalisation and mortality over periods up to 4 years post-assessment in older adults (*Woo, Leung & Morley, 2014*; *Wu et al., 2016*). The SARC-F has reasonable validity and/or reliability when used with older adults across numerous countries and languages (*Bahat et al., 2018*; *Cao et al., 2014*; *Ida et al., 2017*; *Malmstrom et al., 2015*; *Rodriguez-Rejon et al., 2018*). The SARC-F typically demonstrates moderate to good specificity and negative predictive values, but poorer sensitivity and positive predictive values (*Barbosa-Silva et al., 2016*; *Ida et al., 2017*; *Woo, Leung & Morley, 2014*). Such results indicate that the SARC-F may be appropriate at correctly classifying older adults who are not sarcopenic; however, lower levels of sensitivity and positive predictive values indicate it is less able to accurately identify sarcopenic individuals. It is perhaps this poor level of sensitivity that has led the Australian and New Zealand Society for Sarcopenia and Frailty Research (ANZSSFR) to recommend additional research before the SARC-F be recommended for use in clinical practice (*Zanker et al., 2019*).

In addition to its somewhat questionable sensitivity, there appear to be two main gaps within the literature that may impact the SARC-F's potential use in clinical practice and research. The first gap reflects the relative lack of research that has examined the
relationship between total SARC-F scores, individual SARC-F item scores and a range of objective muscle function tests, including those advocated by the EWGSOP2 such as handgrip strength, knee extension strength, chair stand performance and walking speed. The second gap reflects the lack of research that has examined the potential for the SARC-F to accurately quantify changes in sarcopenic diagnosis, muscle function and/or other important health outcomes with proven interventions such as exercise and/or nutritional support. Exercise (particularly progressive resistance training) and nutritional support (e.g., protein supplementation) are being advocated as potential options to reduce the prevalence and severity of reduced physical function/physical disability and sarcopenia in older adults (*Cruz-Jentoft, 2013*; *Denison et al., 2015*). While researchers investigating age-related disability can use a variety of criterion methods to assess muscle function, clinicians working with patients are unlikely to have such time and tools available to them. Therefore, if the SARC-F can detect clinically significant changes in muscle function as a result of exercise and/or nutritional interventions, a variety of medical and allied health researchers/clinicians may have a feasible tool to quantify the effects of these interventions.

Due to the lack of longitudinal data on the SARC-F, the primary aim of this study was to quantify the changes in SARC-F data observed in a step wedge randomised control intervention and their associations with changes in objective measures of muscle function. Secondary aims of this study, which may inform our interpretation of the findings relevant to the primary aim were to use a cross-sectional research approach to: (1) examine how the SARC-F scale at baseline were related to objective measures of muscle function; and (2) explore structure between SARC-F scale items at baseline.

## MATERIALS AND METHODS

### Study design

The data reported in this manuscript describe the results of a sub-study of a single-blind larger trial, where the methods are fully described in a trial protocol (*Keogh et al., 2017*) and within two papers describing different aspects of the overall trial's results (*Hetherington et al., 2018*; *Hetherington et al., in press*). Briefly, community-dwelling older Australians in receipt of government-funded aged care support were recruited to participate in an intervention comprising progressive resistance and balance training (PRBT). In addition to holding an Australian government funded aged care package, the participants were required to be: (1) over 65 years of age; (2) community-dwelling; (3) mobile with or without an aid; (4) able to follow instructions and commit to the study period; and (5) with no recent history (last six months) of resistance training. The exclusion criteria were: (1) requiring more than one person to assist with transfers, standing and/or mobilising; (2) medications and/or diseases with contraindications for exercise; (3) terminal illness or receiving palliative care; (4) an imminent move to residential care; (5) cognitive decline and/or dementia that causes difficult and/or unpredictable behaviours; and (6) inability to obtain a doctor's consent to participate.

All participants were recruited from the membership database of a large north-Brisbane (Queensland, Australia) community and senior citizens centre that offered, among a suite

of other services, aged care support. On behalf of the CEO of the community and senior citizens centre, the research manager sent a letter of invitation to this project to each of its members. A total of 388 individuals replied that they were interested in participating, with 349 identified as eligible by telephone interview. These 349 eligible individuals were sent a study pack containing the participant information sheet, the consent form, health history questionnaire and balance questionnaire and requested to attend the exercise clinic for baseline assessment. Of these, 104 withdrew from the study prior to baseline assessment. The exercise clinic in which the intervention was delivered was also used to perform the baseline assessments. Following the baseline assessment, the 245 participants were randomised by the research manager to exercise (PRBT) or wait list control (CON) at a ratio of 1:2 using block randomisation through a sealed envelope selection method. A modified stepped-wedge randomisation process was used to ensure all participants were provided an opportunity to participate in the exercise intervention if they were interested. The sample of 245 participants was close to the maximum projected 300 older adults who could be recruited to the study and trained in a safe and effective manner (*Keogh et al., 2017*). The stepped-wedge design included four delivery phases of participant entry (between September 2015 and August 2017) that resulted in eight exercise and control clusters. As we aimed to include up to 300 participants in this trial by trial completion at the end of 2017, and working within the constraints of gym size and availability of accredited exercise physiologists, the cluster sizes of the exercise and control groups exhibited some minor variation (*Hetherington et al., 2018*; *Hetherington et al., in press*).

Ethics approval was obtained from the University of Queensland Human Research Ethic Committee (Approval number #2015000879) and the study was registered with the Australian New Zealand Clinical Trials Registry (ACTRN12615001153505). All participants provided written informed consent prior to participation in this study.

## Intervention

The duration of the exercise intervention was 24 weeks. Consistent with previous safe and successful PBRT trials involving older adults with mobility limitations and complex health care requirements as well as current recommendations based on meta-analyses, participants were asked to perform twice-weekly PRBT (*Fien et al., 2016*; *Fisher et al., 2017*; *Grgic et al., 2018*). The resistance training exercises were performed in a traditional manner (i.e., with heavy loads at relatively slow velocities), in an attempt to maximise the muscular strength and to some extent the hypertrophy adaptations (*Reid et al., 2015*; *Schoenfeld et al., 2017*; *Stec et al., 2017*). The balance training exercises included a mixture of static and dynamic balance tasks as recommended to improve walking speed and to reduce the risk of falls and other adverse age-related effects associated with low walking speed and/or low physical activity levels (*Abellan van Kan et al., 2009*; *Granacher et al., 2011*).

Exercise classes involved a light five-minute warm-up, 45 min of PRBT exercises followed by a 5-minute cool down incorporating stretches. All exercise classes involved no more than 10 participants and were performed under the supervision of accredited exercise physiologists who were experienced working with older clients with chronic disease. Resistance exercises were performed on air-pressure driven, computer-integrated

machines proven effective for use among older adults (HUR Australia Pty Ltd., Birkdale, QLD, Australia (*Hewitt et al., 2018*)). The program is overviewed in detail in Fig. S1, and a more detailed overview of the program can be found in the MUAD trial protocol (*Keogh et al., 2017*). If participants experienced pain or discomfort when performing any exercise, the exercise was modified or removed from the participant's program.

Wait list CON participants were requested to continue their regular activities for the initial 24-weeks period. This was evaluated through the completion of a daily dairy. Diaries were completed by all participants during their time in the study, and collected data on sleep integrity, exercise and physical activity engagement, falls and GP, hospital and clinician presentations. These data are outside the scope of this manuscript, but no change in physical activity patterns during the CON phase were observed. To aid retention, the CON group received monthly education seminars covering a range of health and wellness topics. Following the 24-weeks phase, the CON group were given the opportunity to participate in the 24-weeks exercise phase.

## Data collection

A detailed description of the outcome measures collected in the larger trial is provided in the trial protocol (*Keogh et al., 2017*). In summary, data collection occurred at baseline (week 0) and post-intervention (week 25) for all participants enrolled in this study, with the assessors blinded to the participants' allocation. Participants completed the SARC-F self-report questionaire (*Malmstrom & Morley, 2013*). The SARC-F questionnaire comprises five individual items (components) that are each scored from 0 to 2. The first four items assess the older individual with regards to slowness in walking, assistance walking, chair rise and stair climbing ability graded on a scale that represents no (0), some (1) or an inability (2) to perform a given task, respectively. The final question of the SARC-F assesses their falls history in the last year: no falls (0), 1–3 falls (1) and 4 or more falls (2). Total SARC-F scores of four or more are suggestive of sarcopenia (*Malmstrom & Morley, 2013*).

In accordance with the recommendation of the EWGSOP (*Cruz-Jentoft, 2013*), muscle function was assessed by isometric handgrip strength, Isometric knee extension strength, 5-time repeated chair stand and walking speed. While a short description of these methods is provided below, more details can be found in the trial protocol (*Keogh et al., 2017*). Upper body strength was assessed in the handgrip exercise using an isometric Jamar dynamometer (Sammons Preston Roylan, Bolingbrook, IL). Three trials of the dominant hand were performed, with the best trials retained for analysis (*Schaap et al., 2016*). Isometric leg extension strength was assessed with a 0–500 kg strain gauge HUR Performance Recorder (HUR Labs Oy, Tampere, FI) that was positioned on the leg extension machine. With the knee locked at 45°, the participant was requested to push against the machine with maximum force. Participants performed two trials of the isometric knee extension with the best result retained. Physical performance was assessed by the participant's habitual walking speed over 4 m and 5-time repeated chair stands (*Guralnik et al., 1994*; *Studenski et al., 2011*). As per the established protocol, one trial of the 5-time chairs stand was completed and two trials of the walking speed test were performed, with the best trial retained. If

a participant was unable to complete five repetitions of the chair stand test within the maximum time frame of one minute (60 s), they were given a score of 61 s.

Aspects of this wider dataset were used in the present sub-study to describe the demographic characteristics of the sample. These include a variety of anthropometric (body mass, body mass index—BMI), general health (medications and morbidities), function (Activities of Balance Confidence—ABC), mood (Geriatric Depression Scale and Geriatric Anxiety Index) and quality of life (EQ-5D-3L—EuorQoL 5D 3L) measures that have been validated for use in older populations (*Herdman et al., 2011*; *Kurlowicz, 1999*; *Mak, Pang & Mok, 2012*; *Moore et al., 2011*; *Pachana et al., 2007*). A full description of these methods is provided elsewhere (*Keogh et al., 2017*).

## Statistical analysis

Data collected from the SARC-F scale at baseline were used to: (1) explore relationships with objective measures of muscle function; and (2) explore structure between scale items. Linear and multinomial regression models were used to investigate relationships between objective measures of muscle function and SARC-F total score and SARC-F Item scores, respectively. Exploration of underlying structure in SARC-F responses was completed using multiple correspondence analysis (MCA). MCA is a factor analysis technique that is similar to other descriptive methods such as cluster analysis and principal component analysis but operates on categorical data. MCA combines quantitative and graphical methods to assess structure in measurement tools such as the SARC-F that comprise categorical outcomes, and is becoming more widespread in studies involving older adults (*Costa et al., 2013*; *Marcucci et al., 2017*). MCA was selected to describe patterns of responses between the five individual SARC-F items to better understand use of the tool within the specific population. Briefly, the technique assesses deviation from independence between items and identifies relationships in a lower dimensional space that can be visualised. Permissible responses (zero, one or two) for each of the five items comprising the SARC-F questionnaire were plotted with a marker in the lower dimensional space with the first dimension representing the largest deviation from independence and the second dimension (orthogonal to the first) representing the next largest deviation from independence. Interpretation of each dimension was based on relative positions of category markers, with markers closest to the axis and furthest from the origin identified as most important in describing a dimension.

Changes in SARC-F data for each participant included in the four exercise clusters across the intervention were examined for associations with change in objective measures. For multinomial models, pre to post SARC-F items were coded as no-change, positive-change or negative-change. Multinomial models also controlled for baseline response, with $p$-values calculated for regression coefficients using Wald tests. Finally, structural changes in SARC-F items across the intervention were explored with MCA. All statistical analyses were conducted in the R programming language (R Development Core Team, Vienna, Austria) with MCA performed with the FactoMineR package (*Husson et al., 2010*) and multinomial logistic regression models estimated with the nnet package (*Venables & Ripley, 2002*).

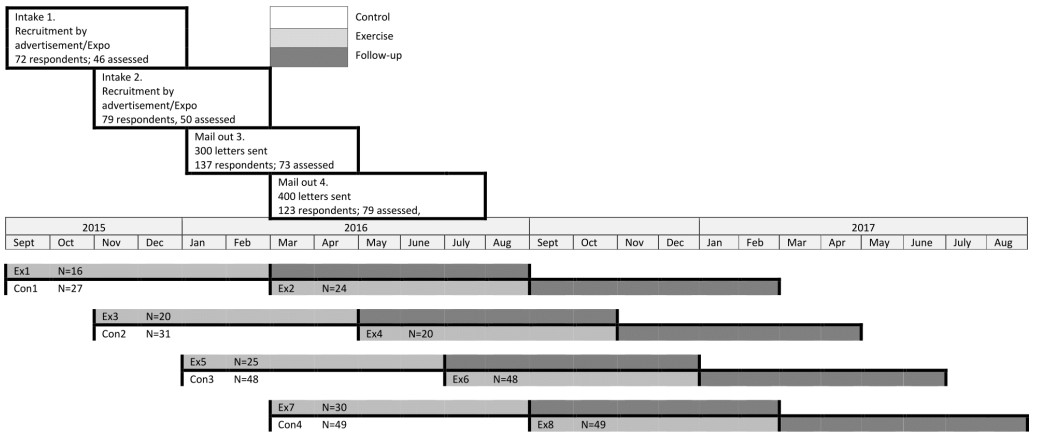

**Figure 1** Participant flow diagram.

**Table 1** Baseline characteristics of participants.

| Measure | Initially recruited cohort ($n = 245$) | Finished exercise program ($n = 168$) |
|---|---|---|
| Age (yrs) | 78.7 ± 6.4 | 78.3 ± 6.3 |
| Gender ($n$, % women) | 195, 80.0% | 131, 78.0% |
| Mass (kg) | 75.9 ± 18.4 | 75.5 ± 17.3 |
| BMI (kg/m$^2$) | 29.4 ± 6.8 | 29.6 ± 6.6 |
| Medications ($n$) | 5.2 ± 3.2 | 5.1 ± 3.1 |
| Morbidities ($n$) | 5.0 ± 2.8 | 4.8 ± 2.6 |
| ABC | 60.7 ± 26.0 | 65.5 ± 24.3 |
| SPPB | 8.0 ± 2.8 | 8.6 ± 2.5 |
| SPPB Balance (s) | 26.0 ± 5.4 | 26.8 ± 5.1 |
| GDS | 3.1 ± 2.4 | 3.0 ± 2.5 |
| GAI | 4.0 ± 4.7 | 3.9 ± 4.6 |
| SARC-F ($n$, % probable sarcopenic) | 85, 34.7% | 34, 20.2% |
| EQ-5D-3L | 0.78 ± 0.15 | 0.79 ± 0.14 |

**Notes.**
All continuous data are presented as mean ± standard deviation. All categorical data are presented as the absolute number followed by the percentage.
ABC, Activity-specific Balance Confidence questionnaire; SPPB, Short Physical performance Battery; GDS, Geriatric Depression Scale; GAI, Geriatric Anxiety Index; EQ-5D-3L, EuorQoL 5D 3L questionnaire.

## RESULTS

A summary of the participants who were randomised to this trial and those who completed the exercise program is provided in Fig. 1. The baseline characteristics of the participants are summarised in Table 1. There were no significant differences between those who completed the exercise intervention compared to those who only attended the initial assessment. At baseline, SARC-F total score demonstrated significant associations (see Table 2) with isometric knee extension strength, walking speed and chair-stand test performance ($p \leq 0.002$). Values presented in Table 2 provide estimates of the relative risk ratio for

**Table 2  Associations between SARC-F sum score or individual item response and objective measures of muscle function.**

| | Isometric knee extension strength | Handgrip strength | Walking speed | Chair-stand test |
|---|---|---|---|---|
| SARC-F Total Score ($r$) | −0.21 (−0.33 to −0.08)[†] | −0.11 (−0.23 to −0.02) | −0.62 (−0.74 to −0.51)[‡] | 0.57 (0.48–0.65)[‡] |
| SARC-F Slowness (RR) | 0.93 (0.62–1.1) | 0.92 (0.69–1.2) | 0.31 (0.20–0.47)[‡] | 2.1 (1.5–2.9)[‡] |
| SARC-F Walking (RR) | 0.73 (0.49–1.1) | 0.89 (0.62–1.3) | 0.27 (0.17–0.42)[‡] | 2.5 (1.8–3.5)[‡] |
| SARC-F Chair (RR) | 0.67 (0.38–1.2) | 0.92 (0.57–1.5) | 0.12 (0.06–0.27)[‡] | 3.5 (2.2–5.5)[‡] |
| SARC-F Stairs (RR) | 0.53 (0.36–0.79)[†] | 0.78 (0.55–1.1) | 0.09 (0.05-0.18)[‡] | 4.9 (3.1–7.8)[‡] |
| SARC-F Falls (RR) | 0.43 (0.19–0.97)[*] | 0.53 (0.27–1.0) | 0.31 (0.15-0.61)[‡] | 1.80 (1.1–3.0)[*] |

**Notes.**

r, Pearson correlation; RR, Relative risk ratio moving from positive to negative SARC-F item response for a one standard deviation increase in objective measure.

[*]$p < 0.05$.
[†]$p < 0.01$.
[‡]$p < 0.001$.

a one standard deviation increase in performance variables responding negatively to a SARC-F item compared with a positive response.

Eigenvalues obtained from MCA with baseline SARC-F data indicated the first 3 dimensions (48.5% cumulative variance) warranted exploration. Results are presented in Figs. 2 and 3 with sarcopenia diagnosis (diagnosed from SARC-F) plotted as a supplementary qualitative variable. Dimension 1 represents overall sarcopenia diagnosis, with negative values for Dimension 1 representing non-sarcopenic individuals and positive values representing sarcopenic individuals. The ability to displace the body vertically (rising from a chair and climbing stairs) appeared to best explain deviation from independence for Dimension 2 (see Fig. 2). In contrast, assistance with walking (walking ability) and falls status items were best represented in Dimension 3 (see Fig. 3).

Descriptive statistics for objective measures and SARC-F items pre- and post-intervention are presented in Table 3. The majority of individuals reported no change in their response for individual SARC-F items (56.5–78.0%) or their sarcopenia diagnosis based on their total SARC-F score (79.2%).

Training-related improvements in SARC-F total score (Table 4) were significantly associated with improvement in walking speed ($p < 0.001$) and chair-stand test performance ($p < 0.001$), but not with isometric knee extension strength ($p = 0.452$) or handgrip strength ($p = 0.878$). Similar results were obtained for multinomial logistic regression models (Table 4), where the training-related changes in walking speed chair-stand test performance were significantly associated with changes in SARC-F items response.

Eigenvalues obtained from MCA with the training-related change in SARC-F items indicated that the first two dimensions (32.2% cumulative variance) warranted exploration. Results of the MCA analysis are presented in Fig. 4 with the training-related change in the SARC-F sarcopenic diagnosis plotted as a supplementary qualitative variable. Dimension 1 is represented by change in SARC-F items, with no change to the left of the origin and change (positive and negative) to the right, with Dimension 2 represented by the direction of the change in SARC-F items. Figure 4 indicates a consistency in response with,
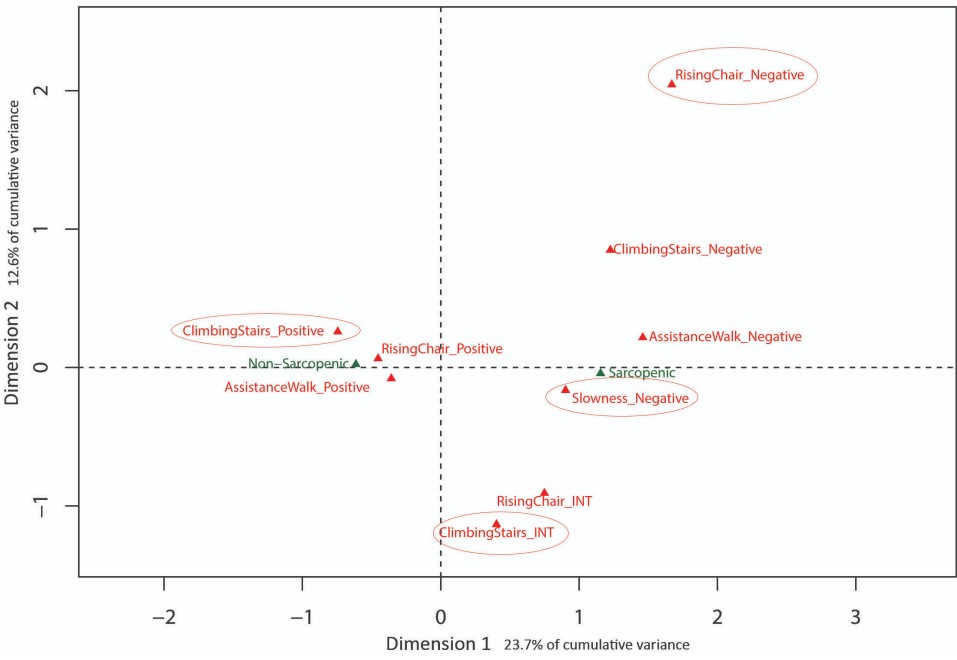

**Figure 2** Graphical representation of Multiple Correspondence Analysis (MCA) of baseline SARC-F responses (Dimension 1 vs. Dimension 2). Factor map identifying most influential categories for dimensions presented along with projected supplementary vectors (sarcopenic and non-sarcopenic).

for example, those experiencing positive change in one item more likely to experience a positive change in another item.

## DISCUSSION

This study demonstrated stronger relationships between the SARC-F total and physical performance measures of chair-stand test performance and walking speed ($r = -0.30$–0.33, respectively) than knee extension or handgrip strength. These relationships were particularly strong for the chair stand test, whereby a one standard deviation reduction in performance resulted in an ~2–5 times increased risk of a negative change in all five SARC-F item scores.

The results of the MCA on the baseline data identified three MCA dimensions that accounted for 48.5% of the cumulative variance of the SARC-F scores, -with the first and most influential dimension separating positive (no difficulty) and negative responses (a lot of difficulty) in completing a task. In contrast, the second dimension was associated with moving the body vertically during chair stands and stair climbing, indicating this ability is influential in explaining variation in item response. Similarly, the third dimension was primarily associated with ambulation, based on differentiating responses related to the ability to walk independently and risk of falls. These last two dimensions of physical function (i.e., ability to successfully move the body vertically or horizontally) have been extensively studied in a variety of older adult cohorts as these tasks appear pivotal in maintaining function, independence and overall health and well-being with increasing age

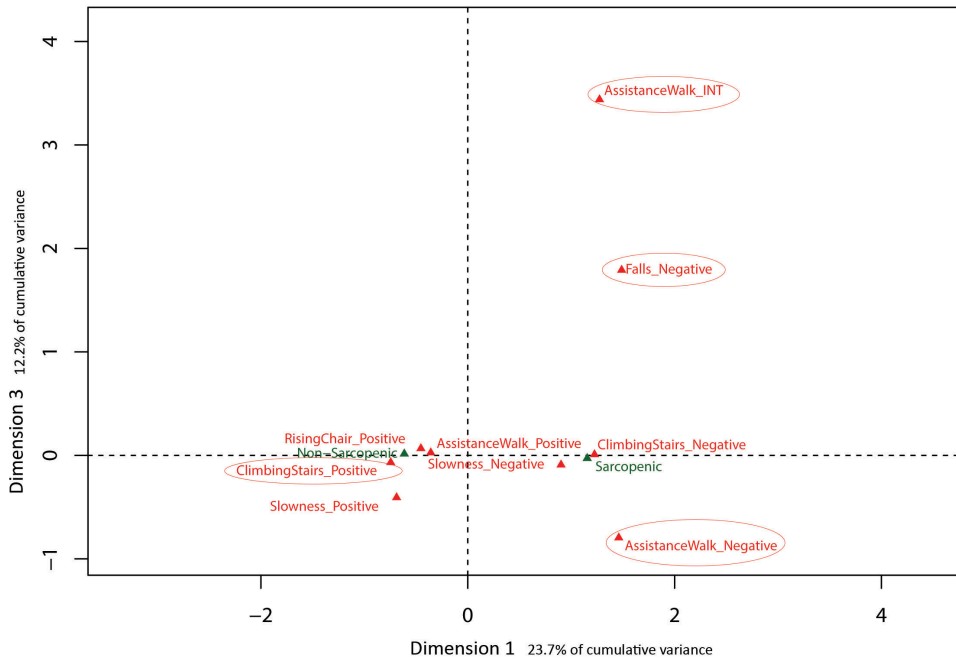

**Figure 3** **Graphical representation of Multiple Correspondence Analysis (MCA) of baseline SARC-F responses (Dimension 1 vs. Dimension 3).** Factor map identifying most influential categories for dimensions presented along with projected supplementary vectors (sarcopenic and non-sarcopenic).

(*Abellan van Kan et al., 2009*; *Bean et al., 2007*; *Howe & Skelton, 2011*; *Keogh et al., 2015*). Collectively, the results of the present study and the literature suggest that the SARC-F may be a useful self-report tool from which older adults can describe their current level of physical function and falls risk.

Examination of the training-related responses indicated significant improvements in the two lower body strength-related outcomes (i.e., the chair stand test and isometric knee extension strength), with trends observed for increased walking speed when compared to the tendency for decline in the control group. Such results appear typical for progressive resistance training interventions, whereby all older adults will significantly increase their muscle strength but exhibit more variable and smaller responses in walking speed and other mobility tasks (*Barbalho et al., 2017*; *Churchward-Venne et al., 2015*; *Fien et al., 2016*).

Even though significant improvements in isometric knee extension strength and chair stand performance were observed after 24 weeks of PRBT, the majority of individuals reported no change in individual SARC-F item scores (56.5–78.0%) or their SARC-F diagnosis (79.2%). This lack of sensitivity to change in the SARC-F scores is consistent with the wider sarcopenia literature, whereby previous studies observed no significant change in sarcopenia (assessed with either the SARC-F or EWGSOP), even though significant improvements in muscle function were observed (*Cruz-Jentoft et al., 2014*; *Hassan et al., 2016*). Another possible explanation for the lack of change in SARC-F items or sarcopenia diagnosis within the current study may reflect the fact that only 34.0% of the sample who completed the exercise program were initially diagnosed by the SARC-F as sarcopenic.

**Table 3  Summary of SARC-F item and muscle function outcomes as a result of participation in the PRBT intervention.**

| Variable | Pre intervention (%) ($n = 245$) | | | Post intervention (%) ($n = 168$) | | | Change pre/post intervention (%) ($n = 168$) | | |
|---|---|---|---|---|---|---|---|---|---|
| SARC-F components | | | | | | | | | |
| | Positive | INT | Negative | Positive | INT | Negative | [+]Change | No Change | [-]Change |
| SARC-F Strength | 45.7 | 19.6 | 34.7 | 54.2 | 20.2 | 25.6 | 29.2 | 56.5 | 14.3 |
| SARC-F Walking | 80.0 | 3.3 | 16.7 | 86.9 | 3.56 | 9.5 | 14.9 | 78.0 | 7.1 |
| SARC-F Chair | 68.6 | 23.3 | 8.1 | 75.6 | 21.4 | 3.0 | 19.0 | 67.9 | 13.1 |
| SARC-F Stairs | 50.6 | 27.8 | 21.6 | 64.9 | 26.2 | 8.9 | 22.0 | 64.0 | 14.3 |
| SARC-F Falls | 60.8 | 33.9 | 5.3 | 66.7 | 29.8 | 3.6 | 15.9 | 70.2 | 13.9 |
| SARC-F Diagnosis | | | | | | | | | |
| | | Sarcopenic | Non-Sarcopenic | | Sarcopenic | Non-Sarcopenic | [+]Change | No Change | [-]Change |
| Sarc-F Diagnosis | | 34.7 | 65.3 | | 20.2 | 79.8 | 13.7 | 79.2 | 7.1 |
| Sarc-F Total Score | | – | – | | – | – | 44.1 | 31.0 | 24.9 |
| Training-related changes | | | | | | | | | |
| | | Pre Intervention (Mean ± SD) | | | Post Intervention (Mean ± SD) | | | Change Pre/Post Intervention (Mean ±SD) | |
| SARC-F Total Score | | 2.5 ± (2.3) | | | 2.0 ± (2.0) | | | −0.5 ± (2.0) | |
| Isometric knee extension strength (kg) | | 6.6 ± (2.9) | | | 9.0 ± (3.2) | | | 2.3 ± (2.0) | |
| Handgrip Strength (kg) | | 20.3 ± (7.8) | | | 21.1 ± (7.5) | | | 0.3 ± (4.0) | |
| Walking speed (m/s) | | 0.89 ± (0.25) | | | 0.91 ± (0.27) | | | 0.02 ± (0.2) | |
| Chair stand test (s) | | 23.1 ± (18.2) | | | 19.1 ± (17.6) | | | −4.0 ± (15.7) | |

Notes.

Positive, No difficulty/0 falls in last year; INT (Intermediate), Some difficulty/1–3 falls in last year; Negative, A lot of difficulty/4 falls in last year; +Change, Item score decreased from pre- to post-intervention/sarcopenic to non-sarcopenic; No change, Item score remained the same pre to post-intervention; Negative change, Item score increased from pre- to post-intervention/non-sarcopenic to sarcopenic.

**Table 4  Associations between changes in SARC-F sum score or individual item responses and change in objective measures of muscle function across intervention.**

| | Isometric strength | Handgrip Strength | Walking speed | Chair-stand test |
|---|---|---|---|---|
| SARC-F Total Score ($r$) | 0.01 (−0.14 to 0.16) | −0.03 (−0.18 to −0.12) | −0.30 (−0.43 to −0.15)[‡] | 0.33 (0.18–0.45)[‡] |
| SARC-F Slowness (RR) | 0.76 (0.46–1.3) | 0.95 (0.57–1.6) | 0.63 (0.37–1.1) | 2.2 (1.2–4.1)[‡] |
| SARC-F Walking (RR) | 1.19 (0.91–3.9) | 1.12 (0.57–2.5) | 0.36 (0.15–0.86)[*] | 2.0 (1.0–3.4)[*] |
| SARC-F Chair (RR) | 1.32 (0.76–2.3) | 0.82 (0.46–1.5) | 0.47 (0.24–0.92)[*] | 2.1 (1.2–3.8)[*] |
| SARC-F Stairs (RR) | 0.83 (0.48–1.4) | 0.57 (0.32–1.0) | 0.40 (0.21–0.76)[†] | 2.0 (1.1–3.5)[*] |
| SARC-F Falls (RR) | 0.92 (0.51–1.7) | 1.13 (0.73–2.4) | 0.94 (0.52–1.7) | 1.11 (0.60–2.0) |

Notes.

$AR^2$, Adjusted $R^2$; RR, Relative risk ratio moving from positive to negative change in SARC-F item response for a one standard deviation increase in objective measure change.

[*]$p < 0.05$.

[†]$p < 0.01$.

[‡]$p < 0.001$.

Collectively, the overall lack of change in the SARC-F items and total score for the PRBT group reinforce some of the limitations regarding its psychometric properties (*Barbosa-Silva et al., 2016*; *Ida et al., 2017*; *Woo, Leung & Morley, 2014*). Linear regression models did however demonstrate that changes in the SARC-F total score were significantly

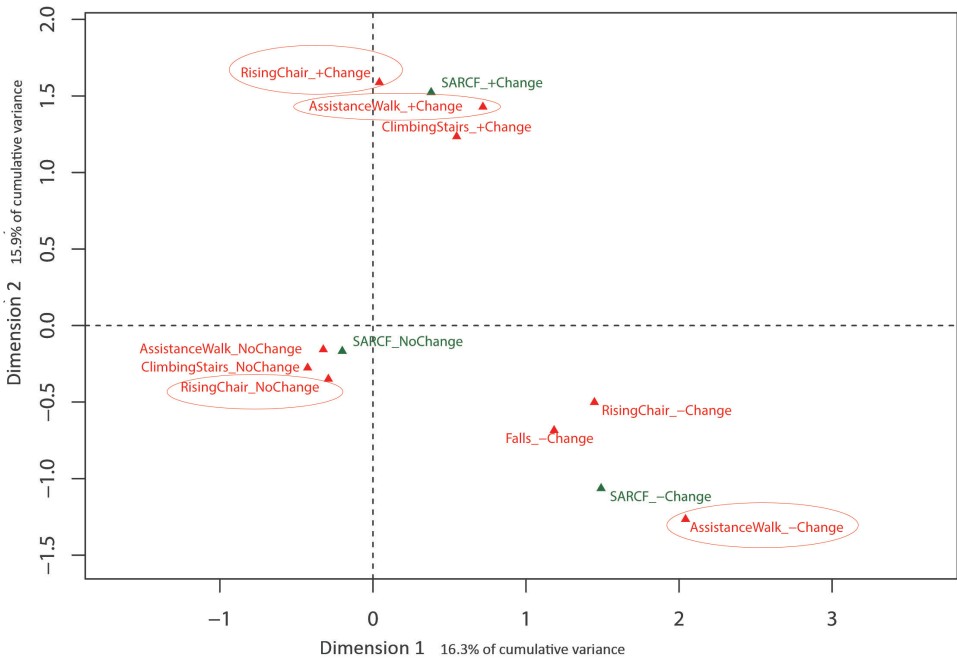

**Figure 4** **Graphical representation of Multiple Correspondence Analysis (MCA) of change in SARC-F responses across intervention (Dimension 1 vs. Dimension 2).** Factor map identifying most influential categories for dimensions presented along with projected supplementary vectors (sarcopenic and non-sarcopenic).

associated with objective changes in walking speed and chair stand test performance ($r^2 = 0.08–0.10$). The strength of these relationships was greatest for chair stand performance, whereby a one standard deviation change (increase or decrease) in objectively measured chair stand performance resulted in close to twice the probability (RR 1.98–2.24) of a one point change in the SARC-F strength, walking, chair rise and stair climbing item scores. Similar results were obtained with multinomial regression models used to account for the categorical nature of individual SARC-F items, where associations between the change in item response were generally associated with changes to in walking speed and chair-stand test performance. Further, MCA results demonstrated that participants who improved performance in one SARC-F item were more likely to also report improved performance in other item(s). Such an outcome supports the importance of lower body muscle strength for an older adult's walking, chair stand and stair climbing performance (*Granacher et al., 2011*).

Our results therefore suggest that older adults may be able to identify clinically significant changes in their muscle function via the SARC-F that correspond to those identified through objective assessments performed by health professionals. This correspondence between health professional derived objective and older adult self-reported subjective muscle function measures is an important finding that many previous studies have been unable to replicate (*Cyarto et al., 2008*; *Liu-Ambrose et al., 2004*; *Van Delden et al., 2013*). Perhaps the inconsistency between our results and the literature reflects the SARC-F's brevity and

simplicity compared to the other tools used in these studies (i.e., Activities of Balance Confidence, Community Balance and Mobility Scale, Stroke Impact Scale—Hand and Motor Activity Log) (*Cyarto et al., 2008*; *Liu-Ambrose et al., 2004*; *Van Delden et al., 2013*).

Future studies should therefore investigate whether the regular use of the SARC-F by older adults may help track their progress and positively contribute to their longer term adherence to the intervention, as older individuals who self-report meaningful benefits from exercise are more likely to continue exercising than those who did not perceive such benefits (*Fisken et al., 2015*; *Hamlin et al., 2016*). Such studies may also look to compare the original SARC-F to a modified SARC-F, as the use of a five item, three-point scale to quantify muscle function and falls risk may not be sufficiently sensitive, with this relative lack of sensitivity likely contributing to possible ceiling and/or floor effects. Currently, one such modified SARC-F has been examined, whereby the addition of the measurement of calf girth demonstrated significantly greater sensitivity in the diagnosis of sarcopenia when compared to the SARC-F alone (*Bahat et al., 2018*). Future research may examine how the inclusion of an additional category per SARC-F item may improve the psychometric properties of the SARC-F. This could be achieved by separating a score of 2 (currently described as "A lot or unable") into two scores (2—"A lot of trouble" or 3—"Unable to perform"). Such a modification may make it easier to demonstrate training-related changes in sarcopenic older adults as each item being would now be scaled from 0 to 3, meaning that the total score would be out of 15 instead of 10.

There was a number of strengths and limitations of the current study. The major strengths were: (1) Its focus on an understudied large older adult population at high risk of entry into residential aged care due to their reduced muscle function and (2) our innovative examination of the SARC-F's relationship before and after long-term PRBT training. The results of this study have implications to the community and aged care sector, whereby an easy to use screening tool for declining muscle function and geriatric syndromes such as sarcopenia will assist a variety of healthcare professionals better identify older adults who need assistance to retain/improve their physical independence and function if they wish to continue living in the community. In contrast, two primary limitations were: (1) that a pragmatic stepped-wedge randomised control trial design was utilised whereby some of the results may be influenced by clustering effects and (2) that the exercise prescription was the same for everyone and restricted to a moderate velocity format.

## CONCLUSIONS

Investigation of the underlying structure of the SARC-F screening tool and its relationship to objective measures of muscle function revealed a number of relevant findings for clinicians. A range of moderate and strong associations were observed with both total SARC-F score and individual SARC-F items, to objective measures of muscle function at baseline and with change across 24 weeks of PRBT. Overall, these findings provide additional insight into the psychometric properties of the SARC-F and provide some preliminary support for how the SARC-F may be used to identify meaningful changes in older adults' muscle function as a result of performing interventions such as resistance training.

## ACKNOWLEDGEMENTS

We acknowledge the contribution of the exercise physiologists and support staff at Healthy Connections Exercise Clinic to the success of this project. We also acknowledge our project partners HUR Australia Pty Ltd., St Vincent's Health Australia, and Burnie Brae. Without Burnie Brae's commitment to health and wellbeing, this project would not have been possible.

### Funding

This work was supported by the Commonwealth of Australia Department of Social Services Aged Care Service Improvement and Healthy Ageing Grant round that closed in 2014 [Grant id: 4-Z35FF5, awarded March 2015] and the Australian National Health and Medical Research Council—Australian Research Council [Dementia Research Development Fellowship to Paul Gardiner #1103311]. The funders had no role in study design, data collection and analysis, decision to publish, or preparation of the manuscript.

### Grant Disclosures

The following grant information was disclosed by the authors:
Commonwealth of Australia Department of Social Services Aged Care Service Improvement and Healthy Ageing: 4-Z35FF5.
Australian National Health and Medical Research Council.
Australian Research Council.
Dementia Research Development Fellowship to Paul Gardiner: #1103311.

### Competing Interests

Dr Justin Keogh is an Academic Editor for PeerJ. Dr Sharon Hetherington and Mr Kevin Rouse are employed by The Chermside Senior Citizens Centre, Burnie Brae, Australia, which was the location of the exercise trial described in the study. Dr Tim Henwood is employed by Southern Cross Care SA and NT. All other authors declare that they have no competing interests.

### Author Contributions

- Justin W.L. Keogh, Tim Henwood, Paul A. Gardiner, Anthony G. Tuckett and Kevin Rouse conceived and designed the experiments, performed the experiments, authored or reviewed drafts of the paper, approved the final draft.
- Sharon Hetherington performed the experiments, authored or reviewed drafts of the paper, approved the final draft.
- Paul Swinton analyzed the data, prepared figures and/or tables, authored or reviewed drafts of the paper, approved the final draft.

### Clinical Trial Ethics

The following information was supplied relating to ethical approvals (i.e., approving body and any reference numbers):

University of Queensland Human Research Ethic Committee approved the study (#
2015000879).

## Data Availability

The data and code are available as Supplemental Files.

## Clinical Trial Registration

The following information was supplied regarding Clinical Trial registration:

ACTRN12615001153505.

## Supplemental Information

Supplemental information for this article can be found online at http://dx.doi.org/10.7717/
peerj.8140#supplemental-information.

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
