# Peer review of "Sarc-F and muscle function in community dwelling adults with aged care service needs: baseline and post-training relationship"

_PeerJ, doi:10.7717/peerj.8140_

## Round 0.1 · original submission · Major Revisions

Dear authors

We have now heard from 2 reviewers on your paper and further actions are need from you. We hope you are willing to perform the revision, sending a rebuttal letter with details and make evident the changes to the manuscript.

·

Basic reporting

Abstract:

1-SARC-F should be added to key words.

Introduction:

2-Introduction is very long and reading becomes a challenge. Should be focused on one thing and expressed in a simple and clear way.
3-Line 92: "As the SARC-F contains five items that… ":The sentence is too long. Should be written in a short and clear way.
4-A study should have a single and primary aim. Otherwise it turns into a very long and complicated article.

Materials and Methods:

5-Line 128: "difficult behaviours": What does it mean?
6-Line 145: "the cluster sizes of the exercise and control groups exhibited were not uniform": Should be clarified.
7-Line 169: "Resistance exercises included: (1) chest press;..." :Those exercises should be presented as supplementary figures.
8-Line 174: "The predicted maximum was a conservatively…": It is difficult to understand. Please write in a simple and clear way.
9-Line 180: "Balance exercises included: (1) single leg stand…": Those exercises should be presented as supplementary figures.
10-Line 200: "A detailed description of the outcome...": It is hard to understand the matter. Please write in a more clear way.
11-Line 207: "The final question of the SARC-F…": A score of zero indicates no falls, a score of two indicating four or more falls, a score of one indicates what? It should be written in order and with all details .
12-Line 227: "and quality of life…": Geriatric depression scale and geriatric anxiety index are used for evaluation of patients’ mood, not their quality of life. Should be expressed differently.


For whole writing:

13-References should be given as numbers in paragraphes and in order; rather than names of the authors of the citated study at the end of the sentence. Names should be given consecutively at the end of the whole writing, under the references title.
14-Please use capital letters only at the start of sentences, out of abbreviations.

Experimental design

Materials and Methods:

1-Line 129: "All participants were recruited from the memberships…": How was the participants reached should be stated clearly.
2-Line 211: "Muscle function was assessed by …": Isometric handgrip strength and isometric knee extension strength show primarily muscle strength, not muscle function. 5-time repeated chair stand test is a power test. These parameters may of course affect muscle function, but they should be categorised seperately from muscle function tests.
3-Line 222: "Due to intense nature of the 5-time repeated chair stand test…": How was it applied? What were the cut-off points? Details should be given.
4-Statistical analysis is too long. It should be shorten and written in a more clear way. Also, multiple correspondence analysis is not a common statistical method. Therefore, some brief and expalanatory information should be given about it.

Validity of the findings

Abstract:

1-Line 35 and 41: "Significant associations…": p values should be written.

Results and Discussion:

2-These parts will be evaluated after they are shortened. Both are very long and hard to understand. Should be written in a clear way.

Reviewer 2 ·

Basic reporting

The article is well written with technically correct and most of clear English. There is an introduction and background to demonstrate how the work fits into the broader field of knowledge.

Line 26 (Abstract) – Consider changing “extend our” to “understand the”. To over detail an explanation may confuse the reader. It’s better to keep it simple and short for a clear message.

Line 31 (Abstract) - Consider mentioning the place of the study, for example by the end of the sentence “… supported aged care in Australia”.

Abstract - It would be better to include information regarding the SARC-F score, such as higher values represent more inability = sarcopenia.

Line 35 and 36 (Abstract); Line 74 (Results) - The correlation between SARC-F total score and measures of lower-body muscle function are not mentioned on the methods and don’t correspond to the results on Table 2 (r= -0.62 to 0.57). It seems chair stand test are not included as lower-body muscle, which is not clear why, but still it wouldn’t correspond to the results on Table 2.

Line 42 (Abstract) – “SARC-F total and item scores and changes in walking speed and chair stand test performance (|r| = 0.30-0.33 and relative risk ratio = 0.40-2.24, respectively).” The 0.30 is the lowest correlation value and it represents an inverse correlation, so a negative sign should precede it (Table 4).

Line 46 and 47 – “These results add to our understanding of the psychometric properties on the SARC-F,”. I recommend a different perspective, as the “… results advance on the comprehension of the psychometric properties...”.

Line 349 and 350 the sentence can be better written, as: “Collectively, the results of the present study and the literature suggest the SARC-F may be a useful self-report tool from which older adults can describe their current level of physical function and falls risk”.

The results are well discussed with recently literature, recognizes it’s on limitations and strengths, and indicates future studies needs.

Line 444 – review grammar spelling.

Experimental design

The authors clearly define the research question, which is relevant. The knowledge gap being investigated is well identified and clearly state how the study contributes to filling the gap. The article also states the chosen statistical analysis, according to the objectives.

It uses appropriate methodology for experimental study to obtain better and effectiveness source for intervention evaluation, as well as better source of scientific evidence.

However, I recommend providing more information about Multiple Correspondence Analysis (MCA). It’s not a common technique used on health research and considering the valuable contribution of the study to clinical use, it’s important that the method provide details to improve the comprehension and interpretation of health professionals about the article results.

Line 227 - I recommend the authors to revisit the literature regarding the definition of quality of life, for example the eudaimonic and hedonic components. Depression and anxiety scales are not measuring quality of life, and the instrument EQ-5D is in fact measuring quality of life related to health.
Steptoe, A., Deaton, A., & Stone, A. A. (2015). Subjective wellbeing, health, and ageing. Lancet, 385(9968), 640-648.
Group, E. (1990). EuroQol a new facility for the measurement of health-related quality of life. Health Policy, 16(3), 199-208.

Validity of the findings

Authors describe in detail the intervention plan with concern for participants' safety, which is especially important for the age group of the study.

It also mentions the research protocol and it provides as supplementary document, which it supports the replication of the study.

I recommend introducing the confidence interval on the regression tables.

The authors could facilitate the graphics interpretation by highlighting/circling the markers mentioned in the results. Despite the methods explanation (Line 46 to 51), considering the complexity of the study, what represent each dimension and the cumulative variance could be mentioned on the graphic legend and on the graphic, respectively.

---

## Round 0.2 · accepted · Accept

Dear authors, thank you for your submission, which I am happy to Accept.

Reviewer 2 ·

Basic reporting

There is great improvement on the revised manuscript.

Experimental design

The authors attended all my suggestions.

Validity of the findings

The authors attended all my suggestions.

Additional comments

The authors attended all my suggestions and the reviewed manuscript shows improvement. The authors were very polite regarding the reply, it demonstrated concern regarding the manuscript's quality and also great knowledge regarding their study.